# Perception of Social and Educational Quality of Life of Minors Diagnosed with Rare Diseases: A Systematic Review and Meta-Analysis

**DOI:** 10.3390/ijerph20020933

**Published:** 2023-01-04

**Authors:** Juan R. Coca, Susana Gómez-Redondo, Alberto Soto-Sánchez, Raquel Lozano-Blasco, Borja Romero-Gonzalez

**Affiliations:** 1Social Research Unit on Health and Rare Diseases and Transdisciplinary Center for Research in Education, Department of Sociology and Social Work, University of Valladolid, 42004 Soria, Spain; 2Social Research Unit on Health and Rare Diseases and Transdisciplinary Center for Research in Education, Department of Pedagogy, University of Valladolid, 42004 Soria, Spain; 3Social Research Unit on Health and Rare Diseases, Department of Psychology, University of Valladolid, 42004 Soria, Spain; 4Psychology and Sociology Department, Faculty of Education, University of Zaragoza, 50009 Zaragoza, Spain

**Keywords:** rare diseases, quality of life, social-educational

## Abstract

This study explores the perception of social and educational quality of life in minors with rare diseases (RDs). Two meta-analyses were performed, applying the random effects model. Results: Regarding the social Quality of Life, the meta-sample consisted of k = 40 samples, with a total population of 1943 children (mean age = 9.42 years), of whom 687 (35.3%) were girls, 615 (31.4%) were boys and 641 (33%) did not report their sex. The effect size was large (mean size = 7.68; *p* < 0.000; 99% Confidence Interval; lower limit = 7.22; upper limit = 8.14). The results of the meta-regression and model analysis showed the importance of the measurement instrument (Paediatric Quality-of-Life Inventory and Prototypes of the Quality of life) and the dissimilarity of perception among caregivers. The nationality and the type of RD were not relevant. With respect to the educational Quality of Life, the meta-sample consisted of k = 19 samples, with 699 minors (mean age = 10.3 years), of whom 266 (38%) were girls, 242 (34.6%) were boys and 191 (27.4%) did not report their sex. The effect size was large (mean size = 7.15; *p* < 0.000; 99% CI; lower limit = 6.35; upper limit = 7.94). The meta-regression and comparison of models showed that the type of RD was essential. The measurement instrument was a moderating variable, especially the Parent version Paediatric Quality-of-Life Inventory. This study reveals the need for further research on RDs and their social–educational effects.

## 1. Introduction

The term rare disease (RD) refers to a large number of conditions with a very low prevalence rate. In fact, in the year 2000, the European Union considered that a condition could be classified as rare when it presents a prevalence rate of 1 out of 2000 people [1]. It has been estimated that 300 to 400 million people worldwide have a RD [2]. Nowadays, there are 6000 to 8000 conditions included in this concept [3]. As was indicated, the notion of RD comprises a very large diversity of conditions. However, all of these conditions share common elements that make them treatable, as a single entity. Many of these conditions generate disabilities and chronic affectations, with delayed diagnoses and around 80% of monogenic conditions [4]. Moreover, most of those affected by a rare disease receive treatments to alleviate their symptoms, since their pathologies cannot be cured [2].

RDs mostly affect the youngest sectors of the population (children and adolescents), with a great emotional impact on those affected by RDs and their families [5]. This is due to the fact that the population affected by RDs requires great care, and the existence of an RD implies a constant management of the disease by the caregivers [6]. Therefore, RDs have an impact on the quality of life of people with RDs and their families. Lenderking et al. [7] concluded that quality of life is an important goal for RD patients. Thus, the challenges described in the mentioned study, according to its authors, are worth the effort and require innovative thinking for each context.

Due to the different biosocial factors, the study of RDs must be multi-disciplinary and multi-stakeholder [8], especially regarding the social dimension (understood in the general sense). In this scope, the analysis of the quality of life provides great information, due to its cross-sectional character. For example, a growing body of evidence suggests that the effects on the quality of life included in the economic evaluations must be considered beyond the individual patient. Therefore, in the present study, we decided to analyse the quality of life from a double perspective: educational and social. However, we performed an analysis based on a stakeholder, after considering the great difficulty in conducting studies on the quality of life in different social groups, since the social factors would be different and the educational factors could even be absent. The aim of this study was the following:


*O: To analyse, through a meta-analysis, the indicators of the quality of social and educational life perceived by the patients and their caregivers, in minors diagnosed with rare diseases.*


## 2. Materials and Methods

The meta-analysis followed the Cochrane Handbook for Systematic Reviews in Higgins and Green [9] and Preferred Reporting Items for Systematic Reviews and Meta-Analyses (PRISMA) [10]. It began with a general search, which was subsequently focused on the social and educational dimensions present in the questionnaires of quality of life. The fundamental reason is the large amount of information obtained, which is why the researchers had to perform a selection of such dimensions. The inclusion and exclusion criteria were established according to the indications of Botella and Sánchez [11] and Moreau and Gamble [12].

### 2.1. Inclusion Criteria

-Sample: Minors diagnosed with RDs aged 0–18 years.Although the paediatric age is up to 14 years, this variable was extended to 18 years to analyse this situation during adolescence and puberty. In this way, the social and educational changes of these stages are contemplated, since they are essential in human development.-Methodology of the articles: empirical and quantitative. Publication date: From 1990 to 2022 (April).-Methodological rigour. Studies of recognised prestige, published in Quartile 1 and Quartile 2 journals (Scimago Journal & Country Rank).-Assessed psychometric instruments present in scientific publications that analyse the quality of life and contain subscales of the quality of social and educational life.

### 2.2. Exclusion Criteria

-Studies with healthy minors that compare these with RD minors and do not specify the experimental and control groups, or other methodological issues that do not allow extracting accurate data about minors with RDs.-Studies that include adolescents and young adults aged 18 years or older.-Specific instruments of quality of life designed for a specific RD.-Studies without quantitative data or with methodological errors [13,14].

The search strategy followed the principles of Botella and Gambara [15] and was conducted in April 2022 in the following databases: ProQuest, Web of Science and Scopus. After successive search strategies, it was found that the Boolean action that best covered the terminology was: “quality of life” AND “rare disease” AND “famil* OR relativ* OR caregiv*”. These searches produced a large number of documents. With the aim of narrowing down the search results, they were limited to the following fields:-ProQuest: “title”, “abstract”, “full text”, “article”, “English”, 1990–2022.-Web Of Science (WOS): “title,” and type of research “article”, 1990–2022.-Scopus: “Article title, abstract, keywords”, “English”, 1990–2022.

The selection of studies was carried out according to the guidelines established in the Cochrane Handbook for Systematic Reviews in Higgins and Green [9] and PRISMA [10], which described the criteria of the studies that make up the meta-sample. Therefore, the first coding step involved a process of systematic and manual review of each study produced by the search strategies. In this way, firstly, each article was reviewed by title and abstract, selecting those with relevance in the topic of RDs and quality of life. Then, the mentioned inclusion and exclusion criteria were applied. Figure 1 shows the flowchart of the present study.

The statistical processing of the data was carried out using two computer programmes. Comprehensive Meta-Analysis (CMA) was used to calculate the effect size (through statistical conversion to Fisher’s Z values), analyse the model comparison and perform meta-regressions and Egger’s bias test. On the other hand, JASP was used to build the graphs, as it allows for a better visualisation of the data.

## 3. Results

### 3.1. Quality of Life: Social Subscale

The sample consisted of a total of 1943 participants distributed in k = 40 samples, in 7 studies [16,17,18,19,20,21,22], of whom 687 were girls, 615 were boys and 641 did not report their sex, with a mean age of 9.42 years. With respect to the geographical distribution, one of the samples was Asian (China), two were European and North American Anglo-Saxon populations (UK, US, Ireland and Canada), and the remaining 37 were European samples (France, Germany, UK, Ireland, Italy and Norway) (see Table 1).

Figure 2 presents the forest plot, which shows a mean effect size of 7.68 (*p* < 0.000; 99% CI; lower limit = 7.22; upper limit = 8.14). In this sense, we found a high mean for the perception of quality of life. As can be observed, the forest plot shows great diversity. However, the analysis of heterogeneity allows determining the degree of diversity. At this point, the analysis of heterogeneity is of particular interest (see Table 2). We have used the DerSimonian and Laird Q indices [23] and the I^2^ for this study. First, the results of the Q index (Q = 2210.115, df = 39, *p* < 0.001) allow the acceptance of the heterogeneity hypothesis. In other words, the meta-sample is diverse. However, it is necessary to ask to what extent this diversity exists. In this sense, the I^2^ index explains how 98.23% of the variability comes from the heterogeneity of the sample and not from chance. This implies that the sample of studies presents varied socio-demographic characteristics, as well as an important variety of procedures and methodologies [24]. These results indicate that the existence of moderating variables is highly probable, which makes it necessary to carry out meta-regressions and model comparison studies [9].

Consequently, we selected a random-effects data-processing model [25,26]. Likewise, Egger’s test was conducted with 99% confidence interval to study the bias effect [11,15]. The results revealed the existence of publication bias with a 99% confidence interval (intercept = 5.30, SE = 2.19; t-value = 2.42; *p*-value 1 = 0.01; *p*-value = 0.02) [27].

In fact, the results of the funnel plot (see Figure 3) show moderate heterogeneity and no significant outliers, indicating the existence of moderating variables. A meta-regression [28] and a model comparison could explain a high variability of results [11]. Before carrying out the comparison between models, we evaluated the statistical significance of each of them, considering only the models with certain significance for the comparison. Thus, models 3, 4 and 5 were excluded from Table 2, as they did not show significant data. The results of the comparison test for each model with the intercept, as a function of the moderating variables, produced the results that are presented in Table 2.

However, only two of them shed light on this aspect based on their effect size and statistical significance. In our results, models 2 and 6 (males and nationality, respectively) explained 0% of the variance. Model 7 (measurement instrument) was very relevant, as it explained 69% of the variance (*p* < 0.000). However, the results of the meta-regression clarify that the instruments PedsQL and QOL-PCD show a differential behaviour (see Table 3).

Similarly, model 8 (informant) was very significant, as it explained 61% of the variance. Regarding the results of the meta-regression (see Table 4), the category of parents and informants whose kinship is not specified presents an adequate statistical confidence (*p*-value < 0.01). On the other hand, the categories patients and mothers were not significant (*p*-value > 0.05).

Although model 9 (type of disease) did not present significance, the results of such meta-regression are provided, since it is a variable in which there is great diversity of results (see Table 5). However, the results of the present study did not show differences in the social subscale of the quality of life as a function of the type of rare disease. Furthermore, Prader Willi’s Syndrome and oesophageal atresia presented poor statistical significance, which should not be discarded until further research is carried out with larger samples.

### 3.2. Quality of Life: Educational L Subscale

The educational subscale in the quality of life of minors diagnosed with rare diseases was found in k = 19 samples, distributed in only three studies, with a total sample of 699 minors (mean age = 10.03 years). Regarding gender, 266 participants were girls, 242 were boys and 191 did not specify their sex. With respect to nationality and culture, we found that all participants were European, specifically from the UK, Ireland, Norway, France and Germany (see Table 6).

Figure 4 (forest plot) shows an effect size with a mean score of 7.15 (*p* < 0.000; 99% CI; lower limit = 6.35; upper limit = 7.94). In this sense, we found a high mean value for the perception of quality of life in the educational aspect in minors with rare diseases.

Figure 5 (funnel plot) shows moderate diversity, which is confirmed by the Q statistic of Der Simonian and Laird [23] (Q = 84.55, df = 18, *p* < 0.001) and I^2^ index, which explains that 78.71% of the variability is due to the heterogeneity of the sample, that is, the diversity of methodologies, procedures and the sociodemographic characteristics of the participants. Therefore, there could be moderating variables that explain such heterogeneity [9,24].

Based on the results of high heterogeneity, we decided to apply a random-effects data-processing model [25,26]. Moreover, Egger’s test indicated the absence of publication bias with a 99% confidence interval (intercept = −0.22, SE = 1.36; t-value = 0.16; *p*-value 1 = 0.43; *p*-value = 0.87) [11,27].

The obtained heterogeneity shows that there could be moderating variables, which makes it necessary to perform model comparison tests and meta-regressions [11,28]. Firstly, we analysed which models were significant, in order to subsequently focus on the comparison between models. The proposed models were: model 2 (males; *p*-value = 0.48), model 3 (females; *p*-value = 0.61), model 4 (nationality; *p*-value = 0.26), model 5 (measurement instrument; *p*-value = 0.04) and model 6 (type of RD; *p*-value = 0.02). Therefore, models 5 and 6 were considered. The results of the model comparison test as a function of the moderating variables show that both models explain most of the variance (see Table 7). Due to the collinearity between some variables, such as the mean age of the minors, the culture of origin and caregivers, it was not possible to analyse these variables.

The model comparison shows that all models were significant (*p* < 0.00). Model 6 (type of RD) explained 51% of the variance, whereas model 5 (measurement instrument) explained 16% of the variance, with strong significance (see Table 7). The instrument PvPEdsQOL presented significant results (coefficient = 0.62; SE= 0.31; Z = 1.99, *p* = 0.04). Regarding the type of RD, this model explained 51% of the variance (*p* = 0.02). However, the results of the meta-regression indicate that the perception of the quality of educational life was significantly lower in the case of Ehlers–Danlos Syndrome (EDS). On the other hand, considering the small number of studies, we cannot discard the significance data of spina bifida/myelomeningocele (MMC) (*p*-value = 0.06), and congenital limb deficiency (CLD) (*p*-value = 0.09) (see Table 8).

## 4. Discussion

### 4.1. Perception of the Quality of Social Life

In recent decades, both the scientific literature and agents involved have emphasised the need to delve into the quality of life of people with RD. According to studies such as that of Boettcher et al. [29], the caregivers of children with RD present a significantly poorer perception of the quality of social life of their children compared to the rest of the informants (control groups, other caregivers, etc.). For obvious reasons, children require thorough research in this sense, in order to detect the alteration in general well-being and quality of life of minors affected by RDs [18]. This entails the involvement of the entire social sector, since children with congenital and infrequent disorders usually need help from many of the parties involved in the public service system [20]. Such supports and facilitators require a high degree of coordination in the interactions aimed at improving the quality of social life.

Further analysis of our meta-sample from the moderating variables shows that it is not possible to establish that sex is a determinant in the perception of the quality of social life. This corroborates the study of Michel et al. [30], who did not detect significant differences when analysing the quality of life in boys and girls. However, given the limitations in the information provided, we cannot assert that sex is not a determining factor. In fact, there are other aspects that must be taken into account. One of these factors could be related to the pro-social character of girls (in childhood and adolescence); another factor could be associated with the more disruptive male behaviour [31,32,33] We know that RDs affect and limit social behaviour; thus, there could be a certain relationship in this regard.

Another moderating variable detected was the measurement instrument. In this sense, we observed that, in the meta-regression, PedsQL and QOL-PCD are significantly differentiated. The consistency and reliability of QOL-PCD are not surprising, since it has been validated for a rare disease such as primary ciliary dyskinesia [16]. These authors stated that the QOL-PCD scale correlates with HRQoL-PedsQL [16]; thus, it is logical that this tool was also very significant in our results. Nevertheless, authors such as Quittner et al. [34] indicate that the use of HRQoL present certain limitations at early ages (2 to 6 years), due to the difficulties in understanding the biomedical language.

A third relevant element that explains the variance of the obtained information is the informant. In this regard, it is important to highlight the difference between the perceptions of the quality of life when it is self-informed by those affected by the disease and when it is informed by the main caregivers, being higher in the former than in the latter [22]. This suggests the possibility that quality of life is greater in the studies based on both caregiver-reported and self-informed instruments—such as the Pediatric Quality-of-Life Inventory (PedsQL) [35] in Bosch et al. [17], Cole et al. [19] Witt et al. [22], Witt and Kolb et al. [36].

This makes sense, since our systematic review was focused on the perception of quality of life, and much of the obtained information was provided by the caregivers of the RD minors. This fact has an impact on the data, as they depend on how the caregivers interpret the reality of the RD minors. Many years ago, Thomasgard and Metz [37] explored so-called vulnerable child syndrome (VCS), according to which the parents usually perceive their children as being more vulnerable to disease and injuries. Based on the meta-regression performed in the present study, we cannot assert that there are significant differences between women and men. However, it has been concluded [33] that women are the main caregivers, confirming what Schweid called the “caring class” [38]. This concept refers to a new social class, in which women play a fundamental role, as they are mostly the ones who provide care. This fact determines the mother–child interactions, and it could be the reason why they report more problems than the fathers [33,39]. However, there are other studies that do not show differences between them [33].

In our case, the meta-regression by informant did not reveal significant data with respect to the mothers. Nevertheless, studies such as that of Chu et al. [40] concluded that, while female caregivers perceive the diseases of their children as highly symptomatic, along with a greater need for biosanitary control of the pathology, male caregivers show more negative perceptions with respect to the general quality of life of the minors. In addition, most of the informants were mothers or female caregivers, which can influence the attainment of greater scores in the perception of the quality of life of minors with RDs [17,21,22,36,41].

In any case, the information gathered in this study must be interpreted cautiously, since, in the category of parents, there is no differentiation by sex; thus, we cannot assert that there is no relationship between perception and sex. In fact, previous studies in adults observed that male adults perceived better quality of life than female adults [42]. Therefore, it would be plausible that the group that shows a more negative perception of the quality of life could also have the same perception toward their children.

In the model comparison, we detected that model 8 (informants) was very important in the explanation of the variance of the results. Our results showed that the perception of the quality of life were not related to the type of RD. In this sense, Uhlenbusch et al. [43] stated that, despite the heterogeneity of the diseases, common life experiences have been found among the families and the patients, including psychological burden, personal and social limitations, interpersonal problems, etc. In fact, Uhlenbusch, Löwe, Härter et al. [44] found that the psychopathology of the type of RD was not associated with its diagnosis, but with cross-sectional aspects linked to the perception of the disease. It would be convenient to add the social elements to these aspects, although our analysis did not detect studies related to the social context; thus, we cannot delve into it. In this sense, Drukker et al. [45] showed that the neighbourhoods of lower socioeconomic level had, in general, lower levels of quality of social life, measured as informal social control, social cohesion and confidence in the children. This could be due to the fact that the quality of life is usually related to psychological rather than social measurement tools.

### 4.2. Perception of the Quality of Life Related to Education

Studies such as that of Cole et al. [19] support the evidence that children with chronic diseases have, in general, worse quality of life than their healthy peers, indicating that it may exert great influence on the barriers to learning and participation. In addition to the medical and social impacts derived from the disease, schooling is very often difficult, interrupted and even impossible [46,47].

In their study with children and adolescents with RDs, families and teachers, Paz-Lourido et al. [48] highlighted that children with RDs are educationally invisible in many cases, which, according to these authors, requires interdisciplinary and intersectoral measures between the healthcare services and the educational institutions.

In our meta-analysis on the perception toward the quality of life related to the educational subscale, it was observed that the models of measurement instrument and type of RD were significant. These results suggest that the instrument to measure the quality of life, as well as the type of RD, are two elements of great relevance in the quantitative study of the perception of quality of life related to education. However, the educational system is a very complex environment, where the elements of interaction and perception also play an essential role. Nevertheless, the existing data are very limited. This hinders the analyses and the making of adequate decisions.

As is reflected in the study of Johansen et al. [20], minors with congenital and infrequent disorders usually need help from many of the parties involved in the public service system. These authors concluded that the effects on the quality of life and the attention to the physical needs derived from the disease must not be in conflict with their socio-educational needs, which implies the need to ensure that the school life of these children is organised in a way that the diagnosis does not interfere with the education and social life of the children more than necessary.

Chu et al. [40] pointed out that caregivers of children affected by RDs considered that they did not have the capacity to take care of themselves or be independent people. Obviously, this affects the perception of caregivers toward the quality of life of children affected by RDs. Although the data we used in the educational subscale are limited, they corroborate the study of Chu et al. [40] regarding the fact that the perceived dimensions of a disease differ as a function of the type of condition. In this sense, the diseases with greater statistical significance are Ehlers–Danlos Syndrome (systemic disease), congenital limb deficiency (embryogenic disease) and spina bifida (neurological and embryogenic disease).

Factors such as a lack of knowledge and the attitudes of school administrations and teachers were not contemplated in the analysed studies. This was explored through a qualitative study, by Paz-Lourido et al. [48], who detected that these elements were fundamental in the quality of life related to the educational environment. For instance, according to Jaeger, Röjvik and Berglund [49], the patients reported that some teachers do not want to adjust the physical tasks to the needs of the student.

For all of the above, further research is expected to allow delving into the effects of the educational system on the quality of life of people with RDs and their families.

## 5. Limitations of the Study

This study presents some limitations related to the lack of rigorous, quality scientific information published in high-impact journals about our object of study. Secondly, as is usual in studies on rare diseases, the sample was limited; thus, the statistical significance of the information is somewhat limited to adequately elucidate whether there are other aspects that could be significant in further meta-analyses.

Finally, a future direction of research could be to analyse these results in two age groups, minors of 14 years old, and above 14 years old, considering countries and health systems.

## 6. Conclusions

The present meta-analysis led us to the following conclusions:The first fundamental conclusion refers to the urgent need of conducting further quality research (quantitative and qualitative) about rare diseases. Currently, the situation of this type of studies is relatively scarce, which hinders the potential of the analysis and the attainment of highly reliable results.The instrument to measure the perception of the quality of life, at both the social and educational levels, is relevant. In this sense, the “instrument” model was very significant at the quantitative level.The informant is also of great relevance when perceiving the quality of life. These data allow us to warn, for future research, about the need of always indicating who the informant is, as well as her/his gender.The perception of the quality of social life is not related, according to our data, to the type of condition. This suggests that many of the people directly or indirectly affected by RDs have similar social needs. Therefore, when making decisions about, e.g., social policies, it would be convenient, in principle, to make such decisions jointly.The perception of the quality of life at the educational level is strongly related to the type of disease. This led us to think about the need of training teachers and schools’ directors in general about rare diseases and developing individualized educational plans for individuals with special needs along with providing the needed rehabilitation therapies in school.

## Figures and Tables

**Figure 1 ijerph-20-00933-f001:**
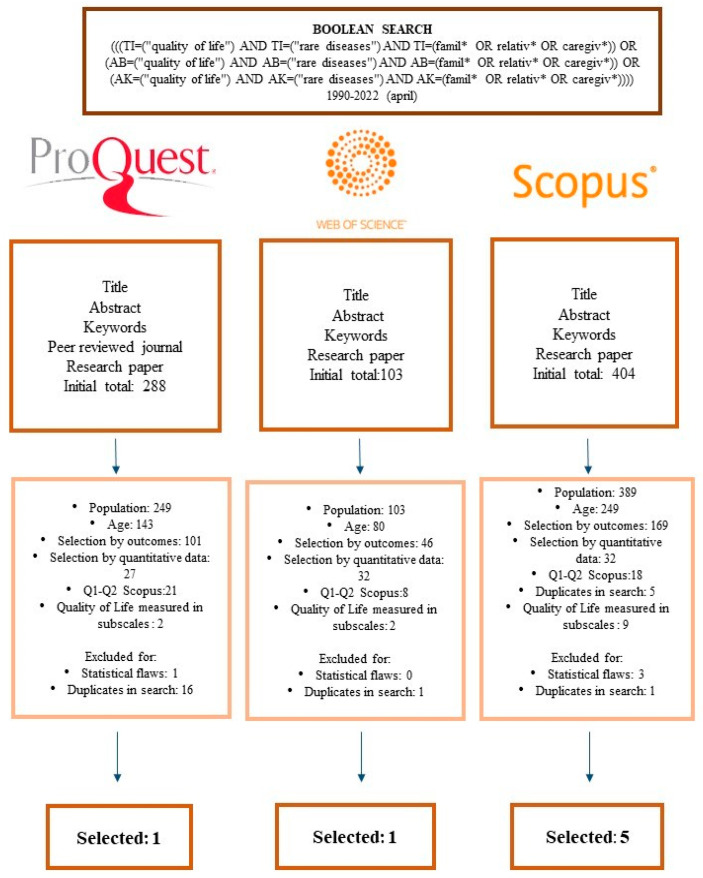
Flowchart.

**Figure 2 ijerph-20-00933-f002:**
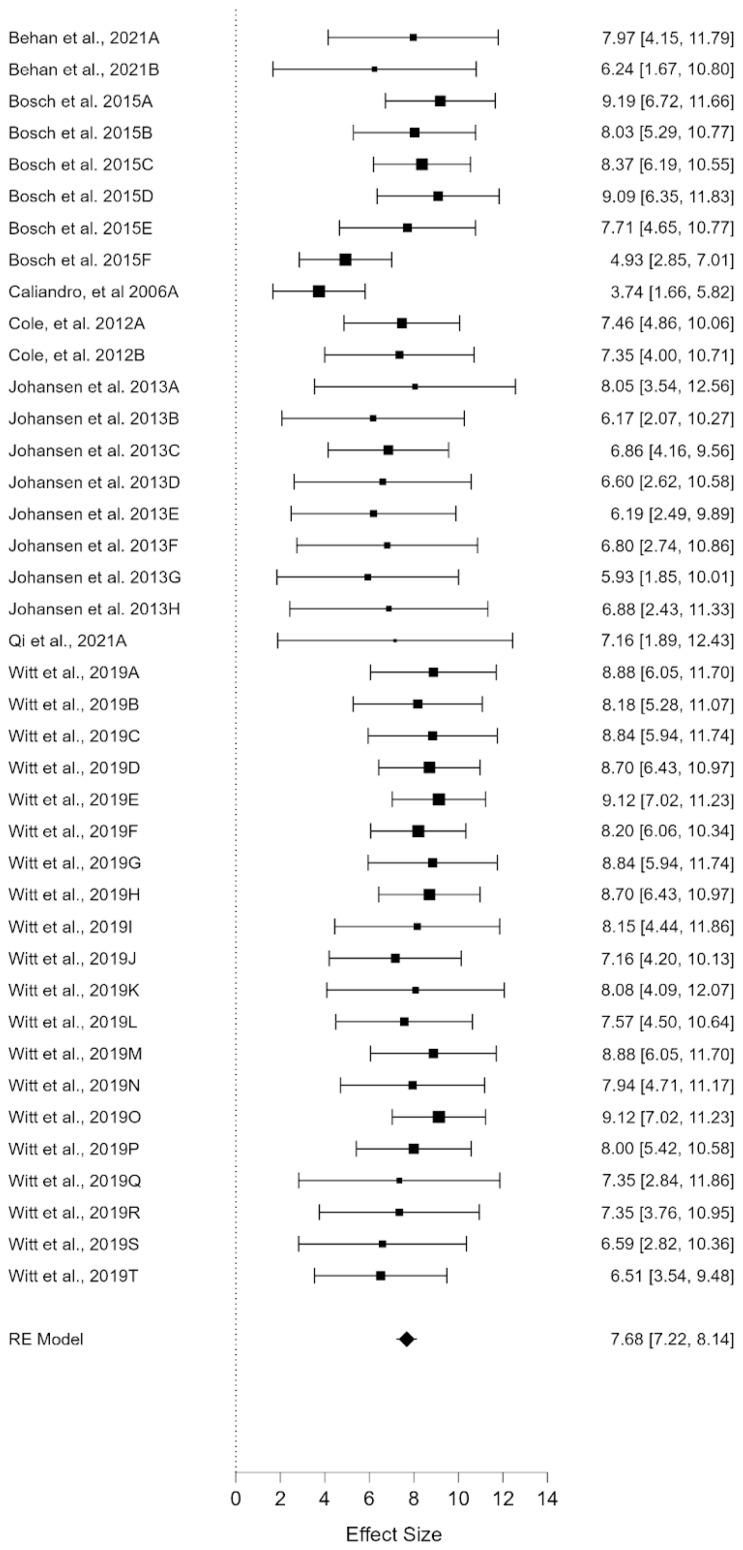
Forest plot of social subscale [16,17,18,19,20,21,22]. Note: Each sample represented on the right-side effect size (lower limit, upper limit).

**Figure 3 ijerph-20-00933-f003:**
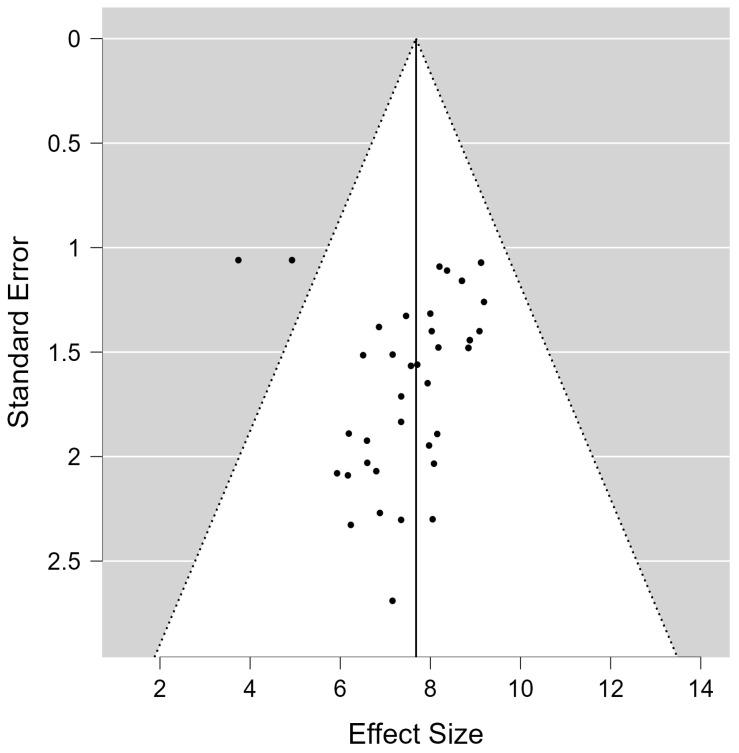
Funnel plot of social subscale.

**Figure 4 ijerph-20-00933-f004:**
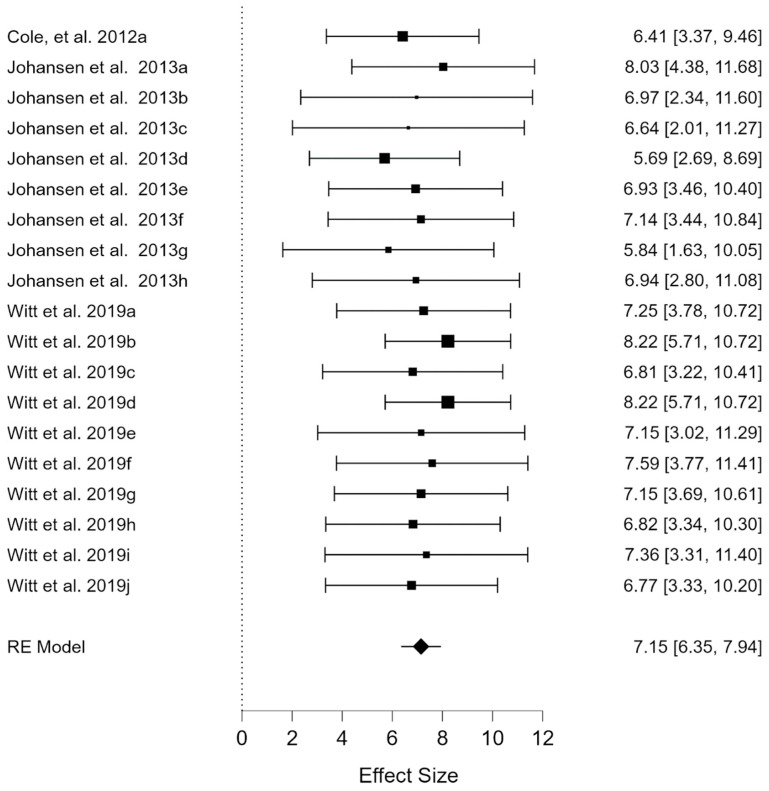
Forest plot of educational subscale [19,20,22]. Note: Each sample represented on the right-side effect size (lower limit, upper limit).

**Figure 5 ijerph-20-00933-f005:**
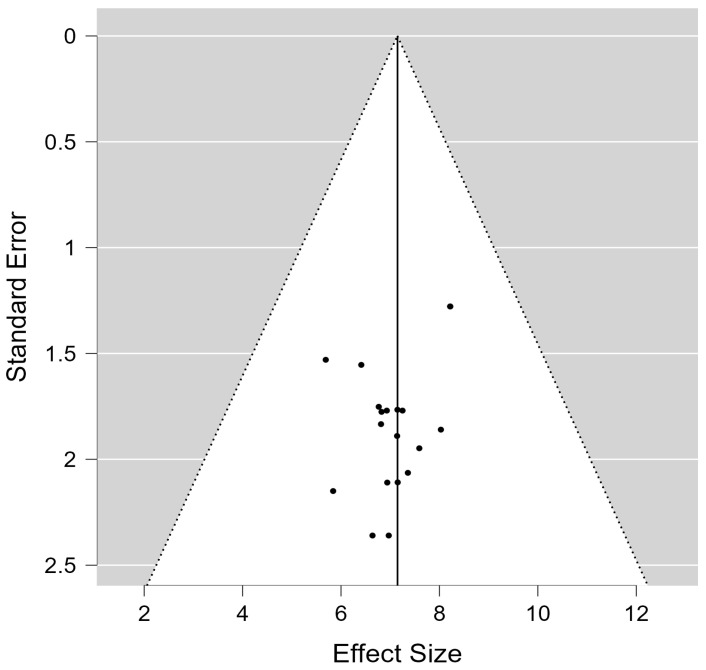
Funnel plot of educational subscale.

**Table 1 ijerph-20-00933-t001:** Sociodemographic results of the sample in the social subscale.

Studies	N	Mean Age	Nationality	Geographic Area	Instrument	Rare Disease
Behan et al., 2021A [16]	71	9.5	United Kingdom, United States of America, Ireland and Canada	Mixed	QOL-PCD	Primary ciliary diskinesia
Behan et al., 2021B [16]	85	15.4	United Kingdom, United States of America, Ireland and Canada	Mixed	QOL-PCD	Primary ciliary diskinesia
Bosch et al., (2015)A [17]	92	9.8	France, Germany, Italy, Holland, Spain, Turkey, United Kingdom	Europe	PedsQL	Phenylketonuria (PKU)
Bosch et al., (2015)B [17]	92	9.8	France, Germany, Italy, Holland, Spain, Turkey, United Kingdom	Europe	PedsQL	Phenylketonuria (PKU)
Bosch et al., (2015)C [17]	92	9.8	France, Germany, Italy, Holland, Spain, Turkey, United Kingdom	Europe	PedsQL	Phenylketonuria (PKU)
Bosch et al., (2015)D [17]	110	14.5	France, Germany, Italy, Holland, Spain, Turkey, United Kingdom	Europe	PedsQL	Phenylketonuria (PKU)
Bosch et al., (2015)E [17]	110	14.5	France, Germany, Italy, Holland, Spain, Turkey, United Kingdom	Europe	PedsQL	Phenylketonuria (PKU)
Bosch et al., (2015)F [17]	253	null	France, Germany, Italy, Holland, Spain, Turkey, United Kingdom	Europe	CHQ-PF28	Phenylketonuria (PKU)
Caliandro et al., (2006)A [18]	9	11.67	Italy	Europe	CHQ-PF56	Prader Willi
Cole et al., (2012)A [19]	17	9	UK, Ireland	Europe	PedsQL	Chronic Granulomatous Disease
Cole et al., (2012)B [19]	17	9	UK, Ireland	Europe	PedsQL	Chronic Granulomatous Disease
Johansen et al., (2013)A [20]	67	11	Norway	Europe	PedsQL	Chronic Granulomatous Disease
Johansen et al., (2013)B [20]	17	12	Norway	Europe	PedsQL	Arthrogryposis multiplex congenita (AMC)
Johansen et al., (2013)C [20]	11	14	Norway	Europe	PedsQL	Marfan’s syndrome (MRF)
Johansen et al., (2013)D [20]	21	13	Norway	Europe	PedsQL	Ehlers–Danlos syndrome (EDS)
Johansen et al., (2013)E [20]	28	10	Norway	Europe	PedsQL	Short stature due to skeletal dysplasia (StSh)
Johansen et al., (2013)F [20]	23	12	Norway	Europe	PedsQL	Osteogenesis imperfect (OI)
Johansen et al., (2013)G [20]	42	11	Norway	Europe	PedsQL	Spina bifida/myelomeningocele (MMC).
Johansen et al., (2013)H [20]	209	14	Norway	Europe	PedsQL	CLD + AMC + MRF + EDS + StSh + OI + MMC
Qi et al., (2021)A [21]	49	7.5	China	Asia	SF-36	Gaucher disease
Witt et al., (2019)A [22]	16	8.03	France	Europe	Pv-PEdsQOL	Oesophageal atresia
Witt et al., (2019)B [22]	16	8.03	France	Europe	Pv-PEdsQOL	Oesophageal atresia
Witt et al., (2019)C [22]	16	8.03	France	Europe	Pv-PEdsQOL	Oesophageal atresia
Witt et al., (2019)D [22]	16	8.03	France	Europe	Pv-PEdsQOL	Oesophageal atresia
Witt et al., (2019)E [22]	16	8.03	France	Europe	Pv-PEdsQOL	Oesophageal atresia
Witt et al., (2019)F [22]	16	8.03	France	Europe	Pv-PEdsQOL	Oesophageal atresia
Witt et al., (2019)G [22]	16	8.03	France	Europe	Pv-PEdsQOL	Oesophageal atresia
Witt et al., (2019)H [22]	16	8.03	France	Europe	Pv-PEdsQOL	Oesophageal atresia
Witt et al., (2019)I [22]	23	8.03	France	Europe	Pv-PEdsQOL	Oesophageal atresia
Witt et al., (2019)J [22]	23	8.03	France	Europe	Pv-PEdsQOL	Oesophageal atresia
Witt et al., (2019)K [22]	23	8.03	France	Europe	Pv-PEdsQOL	Oesophageal atresia
Witt et al., (2019)L [22]	23	8.03	France	Europe	Pv-PEdsQOL	Oesophageal atresia
Witt et al., (2019)M [22]	17	8.03	France	Europe	Pv-PEdsQOL	Oesophageal atresia
Witt et al., (2019)N [22]	17	8.03	France	Europe	Pv-PEdsQOL	Oesophageal atresia
Witt et al., (2019)O [22]	17	8.03	France	Europe	Pv-PEdsQOL	Oesophageal atresia
Witt et al., (2019)P [22]	17	8.03	France	Europe	Pv-PEdsQOL	Oesophageal atresia
Witt et al., (2019)Q [22]	47	9.75	German	Europe	PedsQL	Achondroplasia
Witt et al., (2019)R [22]	47	9.75	German	Europe	PedsQL	Achondroplasia
Witt et al., (2019)S [22]	73	9.75	German	Europe	PedsQL	Achondroplasia
Witt et al., (2019)T [22]	73	9.75	German	Europe	PedsQL	Achondroplasia

Note: The numbering of the samples is shown in alphabetical order, respecting the order of appearance in the article. We chose to name the samples of the social subscale with capital letters, in order to differentiate them from those of the educational subscale, which are presented in lower case letters. All the articles of the meta-sample present multiple samples and/or groups. N = number of participants; QOL-PCD = Prototypes of the Quality of Life; PedsQL = Paediatric Quality-of-Life Inventory; CHQ-PF28 = Child Health Questionnaire version of 28 items; CHQ-PF56 = Child Health Questionnaire version of 56 items; SF-36 = Short Form 36 questionnaire; Pv-PEdsQOL = Parent version of Paediatric Quality-of-Life Inventory

**Table 2 ijerph-20-00933-t002:** Model comparison: Random effects (MM), Z-distribution, Fisher’s Z.

Model Name	TauSq	R²	Q	Df	*p*-Value
Model 1—Simple	1.31	0.00	607.44	20	<0.00
Model 2—Masculine gender	1.31	0.00	607.44	20	<0.00
Model 6—Nationality	2.69	0.00	2210.12	39	<0.00
Model 7—Measurement instrument	2.69	0.69	2210.12	39	<0.00
Model 8—Informant	2.69	0.61	2210.12	39	<0.00

**Table 3 ijerph-20-00933-t003:** Meta-regression by instrument.

Covariate	Coefficient	Standard Error	95% Lower	95% Upper	Z-Value	*p*-Value	VIF
Intercept	4.93	0.91	3.13	6.72	5.4	0	
PedsQL	2.39	0.93	0.55	4.24	2.55	0.01	Q = 37.30. df = 5. *p* = 0.0000
CHQ-PF56	−1.19	1.33	−3.81	1.43	−0.89	0.37	
Pv PEdsQOL	3.47	0.94	1.62	5.32	3.67	0.00	
QOL-PCD	2.17	1.13	−0.03	4.39	1.93	0.05	
SF-36	2.23	1.34	−0.40	4.86	1.66	0.09	

Note: VIF = Variance Inflation Factor; PedsQL = Paediatric Quality-of-Life Inventory; CHQ-PF56 = Child Health Questionnaire version of 56 items; Pv-PEdsQOL = Parent version of Paediatric Quality-of-Life Inventory; QOL-PCD = Prototypes of the Quality of Life; SF-36 = Short Form 36 questionnaire.

**Table 4 ijerph-20-00933-t004:** Meta-regression by informant.

Covariate	Coefficient	Standard Error	95% Lower	95% Upper	Z-Value	*p*-Value	VIF
Intercept	8.36	0.43	7.5054	9.21	19.17	0	Q = 23.89. df = 4. *p* = 0.0001
Mother	−0.16	0.62	−1.384	1.05	−0.27	0.78	
No report	−2.18	0.76	−3.6782	−0.68	−2.86	0.00	
Patients	−0.15	0.53	−1.1998	0.88	−0.3	0.76	
Parents	−1.69	0.52	−2.7328	−0.66	−3.22	0.00	

Note: VIF = Variance Inflation Factor.

**Table 5 ijerph-20-00933-t005:** Meta-regression by type of rare disease.

Covariate	Coefficient	Standard Error	95% Lower	95% Upper	Z-Value	*p*-Value	VIF
Intercept	6.94	0.87	5.23	8.65	7.96	0	
Arthrogryposis multiplex congenita (AMC)	−0.77	1.99	−4.6	3.14	−0.39	0.69	Q = 12.18. df = 14. *p* = 0.59
CLD + AMC + MRF + EDS + StSh + OI + MMC	−0.06	1.93	−3.86	3.73	−0.03	0.97	
Congenital limb deficiency (CLD)	1.10	1.95	−2.72	4.93	0.56	0.57	
Ehlers-Danlos syndrome (EDS)	−0.34	1.98	−4.23	3.54	−0.18	0.86	
Chronic Granulomatous Disease	0.46	1.52	−2.52	3.44	0.3	0.76	
Oesophageal atresia	1.45	0.97	−0.46	3.36	1.49	0.13	
Marfan’s syndrome (MRF)	−0.08	1.97	−3.96	3.78	−0.04	0.96	
Osteogenesis imperfect (OI)	−0.14	1.98	−4.03	3.73	−0.07	0.94	
Phenylketonuria (PKU)	0.93	1.12	−1.26	3.13	0.84	0.40	
Prader Willi	−3.20	1.96	−7.05	0.64	−1.63	0.10	
Primary ciliary diskinesia	0.15	1.50	−2.80	3.11	0.1	0.91	
Short stature due to skeletal dysplasia (StSh)	−0.75	1.96	−4.61	3.09	−0.39	0.70	
Spina bifida/myelomeningocele (MMC).	−1.01	1.96	−4.85	2.82	−0.52	0.60	
Gaucher disease	0.21	1.97	−3.65	4.07	0.11	0.91	

Note: VIF = Variance Inflation Factor.

**Table 6 ijerph-20-00933-t006:** Sociodemographic results of the sample in the educational subscale.

Studies	N	Mean Age	Nationality	Geographic Area	Instrument	Rare Disease
Cole et al., (2012)a [19]	17	9	UK, Ireland	Europe	PedsQL	Chronic Granulomatous Disease
Johansen et al., (2013)a [20]	67	11	Norway	Europe	PedsQL	Congenital limb deficiency (CLD)
Johansen et al., (2013)b [20]	17	12	Norway	Europe	PedsQL	Arthrogryposis multiplex congenita (AMC)
Johansen et al., (2013)c [20]	11	14	Norway	Europe	PedsQL	Marfan’s syndrome (MRF)
Johansen et al., (2013)d [20]	21	13	Norway	Europe	PedsQL	Ehlers-Danlos syndrome (EDS)
Johansen et al., (2013)e [20]	28	10	Norway	Europe	PedsQL	Short stature due to skeletal dysplasia (StSh)
Johansen et al., (2013)f [20]	23	12	Norway	Europe	PedsQL	Osteogenesis imperfect (OI)
Johansen et al., (2013)g [20]	42	11	Norway	Europe	PedsQL	Spina bifida/myelomeningocele (MMC).
Johansen et al., (2013)h [20]	209	15	Norway	Europe	PedsQL	CLD + AMC + MRF + EDS + StSh + OI + MMC
Witt et al., (2019)a [22]	16	8.03	France	Europe	Pv-PEdsQOL	Oesophageal atresia
Witt et al., (2019)b [22]	16	8.03	France	Europe	Pv-PEdsQOL	Oesophageal atresia
Witt et al., (2019)c [22]	16	8.03	France	Europe	Pv-PEdsQOL	Oesophageal atresia
Witt et al., (2019)d [22]	16	8.03	France	Europe	Pv-PEdsQOL	Oesophageal atresia
Witt et al., (2019)e [22]	23	8.03	France	Europe	Pv-PEdsQOL	Oesophageal atresia
Witt et al., (2019)f [22]	23	8.03	France	Europe	Pv-PEdsQOL	Oesophageal atresia
Witt et al., (2019)g [22]	17	8.03	France	Europe	Pv-PEdsQOL	Oesophageal atresia
Witt et al., (2019)h [22]	17	8.03	France	Europe	Pv-PEdsQOL	Oesophageal atresia
Witt et al., (2019)i [22]	47	9.75	German	Europe	PedsQL	Achondroplasia
Witt et al., (2019)j [22]	73	9.75	German	Europe	PedsQL	Achondroplasia

Note: The numbering of the samples is shown in alphabetical order, respecting the order of appearance in the article. We chose to name the samples of the social subscale with capital letters, in order to differentiate them from those of the educational subscale, which are presented in lower case letters. All the articles of the meta-sample present multiple samples and/or groups. N = number of participants; PedsQL = Paediatric Quality-of-Life Inventory; Pv-PEdsQOL = Parent version of Paediatric Quality-of-Life Inventory.

**Table 7 ijerph-20-00933-t007:** Model comparison: Random effects (MM), Z-distribution, Fisher’s Z.

Model Name	TauSq	R²	Q	df	*p*-Value
Model 1 simple (intercept)	0.61	0.00	50.75	9	<0.00
Model 5 measurement instrument	0.61	0.16	84.56	18	<0.00
Model 6 type of rare disease	0.61	0.51	84.56	18	<0.00

**Table 8 ijerph-20-00933-t008:** Meta-regression by rare disease.

Covariate	Coefficient	Standard Error	95% Lower	95% Upper	Z-Value	*p*-Value	VIF
Intercept	7.03	0.35	6.34	7.72	19.94	0	Q = 20.82. df = 10. *p* = 0.02
Arthrogryposis multiplex congenita (AMC)	−0.06	0.79	−1.62	1.49	−0.08	0.93	
CLD + AMC + MRF + EDS + StSh + OI + MMC	−0.09	0.57	−1.22	1.03	−0.16	0.86	
Congenital limb deficiency (CLD)	0.99	0.60	−0.18	2.17	1.66	0.09	
Ehlers-Danlos syndrome (EDS)	−1.34	0.64	−2.61	−0.07	−2.07	0.03	
Chronic Granulomatous Disease	−0.62	0.67	−1.93	0.69	−0.93	0.32	
Oesophageal atresia	0.42	0.40	−0.37	1.23	1.05	0.29	
Marfan’s syndrome (MRF)	−0.39	0.90	−2.16	1.37	−0.44	0.66	
Osteogenesis imperfect (OI)	0.10	0.68	−1.23	1.44	0.15	0.87	
Short stature due to skeletal dysplasia (StSh)	−0.10	0.64	−1.37	1.16	−0.16	0.87	
Spina bifida/myelomeningocele (MMC).	−1.19	0.64	−2.46	0.07	−1.84	0.06	

Note: VIF = Variance Inflation Factor.

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
