# Peer review of "Perception of Social and Educational Quality of Life of Minors Diagnosed with Rare Diseases: A Systematic Review and Meta-Analysis"

_ijerph, 2023, doi:10.3390/ijerph20020933_

Round 1

Reviewer 1 Report

The authors have conducted a review and meta-analysis on the perception of quality of life of minors diagnosed with a rare diseases.

The manuscript is well written, and the interpretation and conclusions are concise.

I have only one major comment, which, however, affects several points:

The manuscript contains a lot of abbreviations, which are quite clear to those readers that are familiar with this kind of mate-analyses. However, I am pretty sure that the manuscript is much easier to read, if all terms are spelled out at the first occurence, with the abbreviation in brackets, as the authors do it for "rare diseases (RD)"

This should be done for all abbreviations:

- QoL, PedsQL, QOL-PCD, HRQOL-PedsQOL in the abstract

Page 2:

-line 62: O1 (could probably be ommitted, since there is no O2 ?)

- line 67: PRISMA

- line 81: Q1 and Q2

Page 4, line 117: CMA

Page 6  line 139 to 142:  the Q index and I2 of the legend to figure 2 must be better explained. It is hard to understand, what the authers mean with the 2 sentences: "Cochrane statistics in Higgins ... and ... the Q index of  DerSimonian ..." (DerSimonian must be written without a blank, by the way. Also in the references")

In addition figure 2 needs a proper heading what the numbers represent (effect size [lower limit, upper limit]

Tables: VIF, etc.

Table 1: It is unclear how the numbers add up. The total is 1,943 as reported in the abstract. However, what is the difference between Bosch et al. (2015)A, B, and C ? As well as several others line that are identicle, but for the  "A", "B", "C", etc. 

Were the probands counted several times ?

Reviewer 2 Report

Kindly see my comments attached.

Reviewer 3 Report

the article entitled "Perception of social and educational quality of life of minors diagnosed with rare diseases: a systematic review and meta-analysis" by Coca and colleagues it is a pretty interesting article regarding the perception of social and educational quality of life in minors with rare diseases by meta-analysis. It is well written and highly recommend to be published after the authors address some of my questions. Thanks.

The article entitled "Perception of social and educational quality of life of minors diagnosed with rare diseases: a systematic review and meta-analysis" by Coca and colleagues it is a pretty interesting article regarding the perception of social and educational quality of life in minors with rare diseases by meta-analysis. I have some doubts to address that may improve the comprehension of the manuscript.

1. First of all, I agree with the authors that we need further research on RD and its social-educational effects. No further action will need

2.- I  agree to include a group of cases between 14-18 years old, but It will be better in a separate way, which may change the results because needs are different for this group. in fact, defining a minor group age could be different depending on the country, and the health system. please consider!

3.- Regarding inclusion criteria in Materials rigour you include Q1 and Q2 articles. Please define the categories used for Q1 and Q2 it is important ( eg. genetics, biology, social...) Depending on the categories used could be more interesting to use 1st and 2nd decile. please discuss!

4.- Please in table 2 model 7 correct the word "insyrument".

5.- in the last paragraph of the introduction section.....For example, a growing body of evidence suggests... Please include the references...

Please address these comments for a better understanding of the MS.
